# Mobile Forensics: Repeatable and Non-Repeatable Technical Assessments

**DOI:** 10.3390/s22187096

**Published:** 2022-09-19

**Authors:** Raffaele Cuomo, Davide D’Agostino, Mario Ianulardo

**Affiliations:** 1Guardia di Finanza di Milano, 20159 Milan, Italy; 2Dipartimento di Informatica, Università degli Studi di Salerno, Via Giovanni Paolo II, 132, 84084 Fisciano, Italy

**Keywords:** mobile forensics, digital analysis, digital evidence, unrepeatability, repeatability, computer forensics, cybercrime, forensic extraction

## Abstract

This paper presents several scenarios where digital evidence can be collected from mobile devices, their legal value keeping untouched. The paper describes a robust methodology for mobile forensics developed through on-field experiences directly gained by the authors over the last 10 years and many real court cases. The results show that mobile forensics, digital analysis of smartphone Android or iOS can be obtained in two ways: on the one hand, data extraction must follow the best practice of the repeatability procedure; on the other hand, the extraction of the data must follow the best practice of the non-repeatability procedure. The laboratory study of the two methods for extracting digital data from mobile phones, for use as evidence in court trials, has shown that the same evidence can be obtained even when the procedure of unavailability of file mining activities has been adopted. Indeed, thanks to laboratory tests, the existence of multiple files frequently and continuously subjected to changes generated by the presence of several hashes found at forensic extractions conducted in very short moments of time (sometimes not exceeding 15 min) has been proven. If, on the other hand, the examination of a device is entrusted to a judicial police officer in order to conduct a forensic analysis to acquire data produced and managed by the user (such as images, audio, video, documents, SMS, MMS, chat conversations, address book content, etc.) we have sufficient grounds to believe that such examination can be organized according to the system of repeatable technical assessments.

## 1. The Acquisition Process of Digital Evidences from Mobile Devices

It should be pointed out that an additional aspect that distinguishes digital investigations conducted in the world of mobile forensics from those similar to those of computer forensics is the absence of free software tools that make it possible to carry out penetrating and effective analyses capable of returning robust evidence from the legal point of view [1].

Although the methodology we present is not based on relevant and innovative scientific results, it must be considered innovative and based on solid foundations because it has been designed exploiting the synergies between the technical component and the legal framework. This means that the procedures that will be presented in the following sections have been considered under two distinct priorities: first from the technical point of view and than for the impact produced on the legal operating framework.

In this work, we aim to highlight the differences that characterize a process for “mobile forensics” with respect to the analogous process that is undertaken in the context of fixed digital devices. This process, according to the standards of ISO-IEC 27037/2012, can be divided into four stages: (1) identification, (2) collection; (3) analysis; (4) preservation and presentation.

Differences are concentrated in particular on stages (2) and stage (3). Thus, it is possible to maintain that best practices in this sector exclude the particular case that the analysis of these devices can be realized in the place where the find was discovered.

The problem is how is it possible to isolate, in the act of collection, the mobile device from the web. There are three main methodologies:Airplane mode (switching off Wi-Fi and any other communication channel such as Bluetooth as well);Faraday cage;Turning on a jammer.

In the case in which it is no longer possible to isolate the device, the only possible option would be to switch the mobile off, to be performed possibly by removing the battery.

It is important to clarify that a further characteristic of digital investigations in the sector of mobile forensics, with respect to the analogous ones in computer forensics, is the absence of free software tools that make it possible to carry out penetrating and effective analyses capable of returning robust evidence from the legal point of view.

## 2. Methodologies for Mobile Forensics

In the context of the digital forensics [1], differences between a process of physical acquisition and the analogous process of logical acquisition are well known: physical acquisition is the only one that is recognized valid from the forensic viewpoint because, in the case in which data are not protected by encryption, this allows to obtain a clone of the acquired support. The latter allows to obtain an invasive analysis and thus to reconstruct not only logical contents, but also the links among single memory locations that have been allocated but not used to reconstruct files and deleted folders.

Logical extraction, instead, is an activity of analysis requiring a greater stock of IT knowledge, since in order to be carried out it necessarily requires the interconnection of the device to a workstation on which there is a software devoted to the backup of the telephone, usually developed by the same manufacturer of the device (we can think for example of “Kies” for Samsung phones using the Android system and “iTunes” for devices manufactured by Apple).

This software reconstructs the contents saved on the device (both internal and removable memory), which are represented both at the file system level (files or folders) and at the application level (organization of contacts, messages, audio, video, images files, etc.) [2].

## 3. The Problem of Unrepeatability

The inspiring element of the technical analysis has been the Legge n.48/2008 derived from the “Budapest Convention” [3] and the body of laws defined in the Italian *Codice di Procedura Penale* (CPP), which describe two methods for acquiring digital evidence:1.Article 359 of the CPP, entitled “Technical consultants of the public prosecutor” provides: “The public prosecutor, when carrying out investigations, reporting, descriptive or photographic surveys and any other technical operation for which specific skills are required, can appoint and make use of consultants, who cannot refuse their opera”;2.Article 360 of the CPP, entitled “Non-repeatable technical investigations” provides: “When the investigations provided for in article 359 of the CPP concern people, things or places whose status is subject to change, the public prosecutor shall notify, without delay, the person under investigation, the person offended by the crime and the defendants of the day, time and place set for the assignment of the assignment and the right to appoint technical consultants”.

Among the different types of extraction that we have presented in the previous section, one needs to select which of them provides the best compromise among multiple factors of different nature, taking into account: non-specialist skills of the judicial police officer who analyses the find; the legal rules governing its operation; the type of technical assessment ordered by the Judicial Authority (i.e., if the assessment is conducted under the constrain of unrepeatable technical assessments then the article 360 [4] of the CPP must be considered; on the other hand, without this constrain (repeatable technical assessments), the article 359 of the CPP [5]) must be applied). In both cases, the goal is to preserve unaltered the device during the investigation [6].

### 3.1. Protection Measures of Accesses to Memory Present in Android E Apple Devices

From a forensic viewpoint, a physical extraction process is clearly preferable to a logical process. Unfortunately, differently from computer forensics, this activity in mobile forensics is particularly complex due to numerous restrictions imposed by proprietary and non-proprietary operating systems installed on smartphones. We can start to analyze this by focusing our attention on the two most widespread operating systems in the mobile phones market: Android and iOS.

With regard to Android, this does not allow full access to the partitions that are present in the internal memory of the telephone, but (the user data partition) it is possible to have access both in reading and writing only to one of them. With regard to the other partitions, the access to the memory is forbidden even for read-only operations, since this operating system, Android, is based on a Linux-type Kernel which, as with all of the countless distributions of operating systems of this type, is designed to be multi-user, i.e., it allows the sharing of resources such that multiple users can share the same machine, since they have access only to those set of resources that have been allocated to them and do not have any possibility of accessing resources owned by other users.

Going back again to the Android system, the considerations we made about the permissions granted to individual users remain perfectly valid, and thus to have full access to the system memory, in order to possibly replicate it in bit-to-bit mode one needs to have root privileges.

In the Linux world, it is not difficult to be granted these privileges since, if the terminal is used by a single user, it is possible from a command line to type specific instructions that allow that first password configuration to be paired with the system administrator and then to start a session of use with these credentials. Unfortunately, in the world of Android systems, this operation is much more complex. Indeed, the only way to implement the so-called system “rooting” consists of modifying some memory partitions listed above, as for example the bootloader and recovery partition.

Even though rooting is strongly discouraged, since a small error during the procedure or in the selection of a custom room that is not suitable to the specific device can lead to very serious consequences with permanent damages to the system, it has also some advantages: for example, a device can become more accessible when it is obsolete and the manufacturer no longer releases updates or when there is the possibility of removing all those applications from the device that have been preinstalled by the producer but are rarely used by the user, with the possibility of freeing up additional areas of previously occupied memory.

The disadvantages, however, are considerable: manufacturers do not consider the act of acquisition of root permits illicit, judging it highly dangerous and detrimental to the normal functioning of the device: the user, after having performed the rooting of the phone, automatically loses the rights associated with the guarantee. Some applications can be designed in such a way that they do not work when on the device detects a manipulation of the bootloader and recovery partition. In general, rooting makes the device less secure, since full access to all system files provides fertile ground for malware that, once installed, can communicate sensitive and confidential data stored on the device to third parties.

After discussing the challenges in order to carry out forensic acquisition from Android devices, we can now discuss the same issues that arise in the context of mobile devices released by Apple and equipped with the iOS operating system. These mobile devices have an even higher protection system compared to the OS Android used by many competitors, since the former are equipped with an encryption technology that, within a few seconds, can make all content in the phone memory inaccessible, only to unlock it as soon as the user enters the unlock code. These measures make it impossible to analyze the device without knowing its unlock passwords and, at present, there are no tools, in mobile forensics, that can overcome these restrictions.

Apart from encryption, iOS also adopts some restrictions that prevents access to the system partition, and even then, one needs to use techniques that allow to remove these restrictions.

In the iOS world, there two principal techniques:(a)Jailbreaking;(b)Use of Device Firmware Update (DFU) mode, i.e., uploading a RAM disk on the device that allows full access to the memory and at the same time using tools to attack the passcode set on the device with brute-forcing techniques.

It should be pointed out that, in recent years, Apple has invested heavily in the study of techniques that would enable rapid encryption of the whole phone: for example, the Iphone 3 model had a hardware component for AES encryption, while the Iphone 4 provided full encryption of system and data partitions. This technique is implemented through a particular organization of the internal memory of the phone: the latter is divided into blocks and, in particular, the first block, called PLOG, is used to store encryption keys and all the data necessary to wiping the device quickly if it is subjected to repeated failed attempts to type the password. There are three encryption keys in the PLOG block and they are called BAGI, Dkey, and EMF!, respectively. The EMF! key is used to encrypt the file system, and each time the device is wiped, the key is discarded and regenerated. If such a key is not available, it is clearly unthinkable to retrieve the contents of the file system. In addition, within the file system, each file is associated with an encryption key: every time the file is deleted, the encryption key is also deleted, so even if, from a physical clone analysis, we can isolate the bit sequence referred to the deleted file, it does not produce any results without a key that helps the decoding operation. Each encryption key is also encrypted with a master key, the complexity of which depends on the security class. The iOS provides several classes of protection, each identified by a master key. When the user types the passcode, it actually acts on the two master keys combined with the two main security classes, thus discovering the files encrypted with the keys encrypted with the master keys unlocked.

There are also files within file systems that are matched with encryption keys that do not match any master keys, or any security class. The encryption keys of these files are stored in the Dkey block of the internal memory.

This is why it is necessary to boot the device with a RAM disk, since doing so would enable the analyzer to access the keys stored in these special memory blocks, especially the EMF! block and the Dkey block. Until version 5 of the Iphone, the passcode unlocked data such as e-mail messages, internal files containing the passwords of wi-fi networks that were hooked up by the device and for which an automatic connection was set, and data files from third-party applications. These data, especially in the context of a digital forensics survey, are indispensable, but without knowledge of the passcode, unfortunately, it is inaccessible. This explains why, as far as I-Devices are concerned, it is necessary to try to circumvent the security measures present in the system in order to conduct an effective forensic analysis. Booting the device with a carefully assembled RAM disk for this purpose allows, on the one hand, access to the blocks of memory where the encryption keys are stored, and, on the other hand, allows, with brute-forcing, to attack the passcode without incurring the risk of activating the wiping of the phone following repeated insertion of a wrong unlocking code.

The second technique that can be used for forensic analysis is jailbreaking. This is the procedure that removes software restrictions imposed by Apple on iOS TvOS devices. It allows us to install third-party software and packages, unsigned and authorized by Apple, as an alternative to those of the App Store. After jailbreaking the device, you can install many applications or modify the software through alternative stores such as Cydia, Icy, Rock and Installer, on the understanding that a “jailbroken” device still can launch, and update applications bought from the official store.

### 3.2. The Constraint of Repeatability

The previous section has allowed us to understand that, for both Apple and Android family devices, although physical extraction is the one that most closely adheres to the legal constraints of any forensic analysis activity, the same extraction encounters many difficulties, unlike in the case of extraction conducted in the computer forensics. While full access to memory is indispensable for a comprehensive and legally effective analysis, it appears difficult to implement for smartphones.

The considerations made thus far tend to give less weight to the hypothesis that the process of acquiring digital evidence from smartphones can be considered an act of a repeatable nature [7]: not only is there a need to work on the phone, but more importantly, internal firmware alterations must be made that compromise the operation of the device as displayed before it is seized from the owner. Here we will try to put forward a different position, more in line with the needs of the judicial police, as to whether or not the acquisition process can be repeated [8].

Let us start by saying that the reflections made so far about how to achieve a complete acquisition of the phone concern only techniques of common use, that is, those allocated to those who, although not equipped with specific instrumentation, intend to make a “full disk” acquisition of the internal memory chip of a phone. These limitations do not concern the work of the judicial police, which in any case has expensive instruments, certified by international standards and designed for this type of technical inspection. Certainly, we have repeatedly suggested caution, reiterating that the belief that the judicial police have “secret” tools and techniques that can overcome any obstacles and measures of protection in the computer field is wrong (a cyber investigator, for example, can only take note of the existence of an encrypted volume but does not possess techniques other than those of brute-forcing to try to discover the reserved word). With regard to the technical inspections carried out on devices of that nature, it is quite impossible to assume that a physical acquisition is an act of an unrepeatable nature for any device existing on the market. For some devices, in particular older devices, it is possible to carry out acquisition operations starting from a phone-off condition. Some instruments are able to physically acquire certain Samsung smartphone models from a switched-off phone situation that is, however, initiated through an external impulse, in a particular condition of operation. “download mode” (analogous to computer forensics techniques where the boot of the system was launched from a live Linux distribution).

In any case, regardless of whether or not it is possible to carry out a physical acquisition on any device capable of returning a computer assessment and a digital test in a perfectly repeatable manner, we consider it appropriate in this section to set out some considerations on whether or not to carry out such an operation on devices of this nature. In their opinion, the protocols and “best practices” followed in the world of computer testing should not obey only legal constraints.

Rather, the inspection must be more influenced by technical constraints, which are directly linked to the characteristics of the device under investigation. It is obvious that the majority of legal disputes and criminal proceedings will depend on the results of expert reports or inspections carried out on IT devices and that the techniques used will be constantly evolving and will always have different characteristics, with the result that it is impossible to provide for methods and processes for such investigations that guarantee the return of digital evidence collected always and in any event in a perfectly repeatable manner. Without going far away from contemporaneity, we imagine that in a general criminal case, a suspect confesses their guilt, stating that on a cloud of their possession he retains numerous digital evidence that would facilitate the work of the judicial police in the phase of searching for the sources of evidence, in exchange for a possible reduction in sentence. Clearly, we immediately understand that in such a scenario it is impossible to attempt a physical acquisition process, not only because the server hosting such data may be in a foreign country not available to receive international requests, but also because the server hosting the data cannot be shut down to carry out the acquisition in bit-stream of the internal memory (assuming that it has such dimensions and characteristics that it concludes the process in a reasonable time). That is why the only way to find such kind of evidence is to download it and, if so, it would be necessary to address further difficulties falling within the scope of network forensics, together with the fact that the Internet, being a “best-effort” infrastructure, offers no guarantee as to the order in which the bits assembling the data are downloaded. Most likely, the files downloaded from time to time will always have the same hash, but it cannot be established with certainty that the same download, repeated countless times, always returns a file with the same hash code.

This simple example is considered here to show that the physical extraction process, although more adherent to legal constraints and aspects, is nevertheless an operating methodology successfully adopted exclusively in the computer forensics world, but the chances that the same will be successfully adopted in any future scenario are extremely low. Moreover, returning to the technical computer investigations conducted on mobile devices, a total acquisition of internal memory is an inefficient operation because it is carried out with considerable effort (especially with high risks of causing a “brick” of the phone) to collect a quantity of data of which only a small part is actually interesting for the investigations. The data of interest to the investigator coincide, in almost all the investigative scenarios, exactly with those produced by the user, as the only ones able to reliably reproduce details relating to frequencies, social relations, messaging exchanges, and so on. The only constraint that the investigator has to follow in order to turn these data into valid digital evidence is not to alter it either during the extraction or during the custody, which clearly does not occur in case of a logical acquisition and is excluded in case of a physical acquisition that requires a preliminary alteration of the device firmware or a switch on and off of the phone during the same procedure where only system files and not user data would be touched.

Moreover, in the case of devices capable of interacting with the Internet, it should be noted that Legislative Decree No 109/2008 has stipulated precise timelines regarding data retention by providers providing electronic and telematic communications services. This rule is a great advantage for digital investigators, since, especially in the case of telephones, there is the possibility of conducting analyses at a more superficial level, since the ‘traces’ that the telephone has left in the network (locked cells, telephone calls, telephone identifiers in which a certain SIM has been inserted) represent information that can be easily found by means of a request made by the Judicial Authority to the telephone operator used by the suspect, so that a physical extraction would return an excessive amount of data, of which only a small part, perfectly repeatable, would actually be useful for investigations.

Of course, all the considerations made so far have one common factor: it is not possible to conduct a physical extraction that is perfectly repeatable, meaning by this the possibility to obtain, from time to time, a clone of the memory from which the same hash is extrapolated. The extraction of digital evidence that is actually of interest for investigation purposes (data within the ‘user’ data partition) is a perfectly repeatable operation, since there is with absolute certainty an independence between the internal state of the analyzed device and the data contained in the user partition [1].

Our inferences lack legal confirmation, i.e., an “imprimatur” on the part of the legislature, which also accepts our conclusions from the point of view of case law. Indeed, as can be seen with regard to Law No 48/2008, the legislature did not discuss technical matters but merely outlined aspects relating to the data collected and presented in the debate, and did not specify any technicalities: this is not only because of the sensitivity of the subject being addressed, but also because the computer world is constantly evolving. In reality, what would benefit the digital investigator is not the introduction of well-defined best practices by lawyers governing the way in which the computer search is carried out on any telephone (which is obviously impossible), but rather taking a position according to which the computer data, intended to become digital evidence, must not be assessed on the basis of the techniques used for its collection, but on the basis of its ability to represent evidence and the possibility of not being subject to unintended alterations caused by the device on which it lies. In other words, if the data always has the same evidential content and is of such a nature that it is not altered, not even accidentally, so that it can always be associated with it and in any event a unique hash code, then the principle would prevail that the evaluation of the digital evidence should not be made from a purely formal point of view (valid only if it comes from a physical extraction process that enjoys full repeatability characteristics) but, rather, from a strictly substantive point of view (the data keeps equal hash code and equal evidential content), so that the acquisition techniques adopted would pass second level, to the advantage of an evaluation made exclusively on the data of interest [9].

## 4. Analytical Methodologies Used

Before discussing the study of the repeatability of the extraction process, we thought it appropriate to first describe in detail the techniques and tools used to conduct the acquisitions [10].

The most important points to be resolved are:1.One must be able to compute the hash value of each file that is obtained from a acquisition process;2.A methodology must be developed, which makes it possible to compare, for each file extracted from the same phone with several different acquisition processes, the hash value it presents from time to time. Both issues clearly require automatic solutions, as the number of files expected to be obtained in each extraction is certainly not small [11].

As regards the tools used for extraction, the forensic acquisition software UFED4PC v.7.12.0.14 provided to the “Computer Crimes Team” at the Public Prosecutor’s Office of the Republic in Milan has been used. That software, in essence, has the same interface as UFED Touch, except that in this case the acquisition can be carried out by directly connecting the phone to the personal computer on which UFED4PC is installed. Once the acquisition process is complete, UFED4PC produces a set of files, depending on which extraction technique is performed. The types are as follows:Image file in case of a physical acquisition process;Compressed files (.zip) in the case of a file system acquisition process;Media files and backup files in case of a logical acquisition process.

Each acquisition process, independently of what it is, also produces a file in proprietary format, with the file extension .ufd or .ufdx. This file can only be read through UFED Physical Analyzer. Once the above files have been opened through the Physical Analyzer, it is possible to ‘download’ them, i.e., processing the file obtained downstream of the acquisition process in order to extrapolate from the file found on the phone. These files are also subdivided by type and stored in a folder (for example, the “Image” folder which will contain the extracted images; the “Audio” folder for audio files, etc.). Together with the extrapolated files, an executable file is also generated, a file signed with the title “UfedReader.exe” that, once launched, allows access to a rich informative heritage with the possibility also to generate a wide report in numerous formats (.xls, .pdf, .html, etc.). Once one obtains the files extracted from the phone under investigation, one needs to compute the hash of each individual file. UFED4PC is also useful in this case, as the reports that are generated allow to compute, for certain types of files, as many as two hash keys produced following the MD5 algorithm and the SHA-256 bit algorithm. In addition, the report allows to organize, in an interface of immediate understanding, also data of other nature, generating, for example, the list of incoming and outgoing phone calls, the list of text messages exchanged, the messaging conversations realized and more.

The main limitation of this solution lies precisely in the fact that hashes are computed only and exclusively of files that may be of interest for investigative purposes, leaving out files of other nature instead. To remedy this problem, it was decided to use the commands available under Linux and usable through the shell. Specifically, we used two commands: the first is md5deep with the recursive option; the second is the sha1deep with the same option. In other words, through the Linux shell, selecting the folder within which all the files obtained by the acquisition process are nested, including also the reporting files (these clearly not of interest for our purposes), we launched the two commands described above which, thanks to the recursiveness option, have computed, starting from a root node (folder), all the hashes of all the files contained in all the nodes (folders) leaf. The computed hashes have been collected, using the output management options for the results, into different text files, following a format that shows, on each line, the path of the file and its hash code. The data thus collected were subsequently transferred to excel sheets specifically created for these experiments and were cross-referenced using automated tools.

## 5. Repeatability Analysis of Acquisition Processes Conducted on a Samsung Galaxy S4-I9515 Smartphone

One of the phones tested is a Samsung-branded smartphone, Galaxy S4-I9515 with Android operating system version 5.0.1 and an internal memory capacity of 16 GB. Before starting the tests, the phone was properly configured so that the analysis of activities could be carried out without any hassle and all protection measures, consisting of unlock codes or pattern locks, were removed. The operations conducted on the phone are exactly the same as those suggested by the UFED4PC software, namely:Disabling the automatic shutdown system;Entering airplane mode;Setting the Media Transfer Protocol (MTP) as communication protocol for interfacing the phone with the workstation via an USB connection.Developer mode activation;

After these configurations were made, the device was attached via a USB cable to the location where UFED4PC was installed. The software automatically recognized the device, so the analysis could start immediately. The first transaction was a physical acquisition activity. In addition, following the suggestions provided by the acquisition software, the phone was first switched off, then the yellow plug “T-133”, supplied to UFED4PC, was hooked to it. The phone therefore started in download mode, after which it was connected via USB cable to the terminal used for acquisition. After a preliminary reading of information, the capture software required a disconnection of the device, a switch on in normal mode, a switch off, a reconnection of the aforementioned yellow plug to the phone as a result of which the phone started again in download mode. At this point, the phone, always holding the yellow plug attached to it, was connected back to the acquisition computer and the extraction process started regularly. The task lasted a total of one hour and 46 min and two files were generated at the end of the task:Samsung GSM_GT-I9515 Galaxy S4.ufd, size 1 KB;Dumpdata.bin, image file corresponding to the physical cloning of the phone under consideration, size 15,388,672 KB;

The two files were saved in a folder named “Physics_1”. At this point we generated through the aforementioned file with extension .ufd the reporting related to the acquisition just ended. The files assembled from the image produced were all saved in a folder called “Report”, also contained in the aforementioned folder “Physics_1”. After this first acquisition, we performed a second data extraction from the phone, always choosing a physical acquisition activity. The extraction activity lasted a total of one hour and 44 min, at the end of which we obtained two files with name and size in KB perfectly coinciding with the analogous files obtained in the previous case. These data have been saved in a folder called “Physics_2”. Again, by using the .ufd file extension, we generated the reports and files obtained from the extraction process: these files were saved in a folder called ‘Report’, which is also contained within the aforementioned directory ‘Physics_2’. We then went to file system extraction. For this activity the phone was kept on, remaining always in airplane mode, and no different settings were made than those already set for physical acquisitions. The file system extraction activity lasted 11 min and 38 s and produced the following files:Samsung GSM_GT-I9515 Galaxy S4.ufd, size 1 KB;Samsung GSM_GT-I9515 Galaxy S4.zip, containing the extracted data, of size 42,697 KB.

As with previous acquisitions, these data were saved in a folder called “File_system_1” in which we also saved the report obtained through the file with the extension .ufd. Following this acquisition, we conducted another identical one, lasting 11 min and 28 s, which produced the same files as in the previous case. They were saved in the folder marked “File_system_2” where we also downloaded the files obtained from the report. Finally, two logical acquisition activities were carried out. Again, no different phone configurations were required from previous acquisitions. The first logical extraction lasted 8 min and 21 s. This procedure, being rather similar to a normal backup, is able to return for the most part multimedia files produced directly by the user. In addition, “container” files are also generated, in which content such as the call log is saved; all incoming and outgoing text messages stored on the telephone; all conversations via messaging apps; the set of contacts in the address book, etc. In this case, again, clearly, reporting files are produced in HTML format, containing an illustration of the list of the various types of data found on the device.

This is not to be confused with a similar report generated through the .ufd file using the Ufed Physical Analyzer program. The set of files obtained from the logical acquisition were saved in a directory marked with the name “Logica_1” and subsequently those of reporting were saved in a folder inside the latter, in a manner entirely analogous to the previous cases. Later, a second logical acquisition was made, also of the same duration as the previous one, whose files were saved in a folder called “Logica_2” in which we also downloaded the reporting files. Once we have obtained the data of the single acquisition processes, we have computed for each of them its hash code, both with the MD5 algorithm and SHA-1, and then moved on to the subsequent comparisons thanks to the macro-internal excel files described above.

Let us first point out that it is beyond any rationale that a single phone undergoes repeated acquisition processes: the police officer will assess whether only a single acquisition can be made on that device, clearly choosing the one that can produce the most information that best meets the investigative needs.

To have a complete study on the repeatability or otherwise of the capture process, we chose to do more data mining to compare the obtained results. In any case the comparison of results between operations of different types is not easy, as the data obtained for each type of extraction will be different, both in quantity and type; therefore, since equal extractions will produce the same number of files, while different extractions will produce a different amount of files, we have decided to adopt the following criteria for comparing the data: for the first type of comparison, we decided to use the hashes obtained directly from the commands typed in the Linux shell, while for the second type we used the hashes contained in the reporting files automatically generated by the acquisition software. Independently on the acquisition processes adopted (physical, logical, or file system) the files we have found fall in one of these categories: (a) database files; (b) image files; (c) text file.

### 5.1. Comparing Physical Acquisition vs. Physical Acquisition

Let us start by comparing the results obtained in the two physical acquisition processes. The two acquisition processes generated two files respectively, one of which, the Dumpdata.bin file, is the physical image obtained from the internal memory. Before comparing the hashes of the various files, we found it interesting to also compare the hashes of the two physical images produced. As shown in the figure, the hashes of the two images do not coincide, but this aspect is neither a surprise nor an anomaly, since as we have already specified, between the two physical acquisitions the phone has been restarted, which is why the internal state (and therefore the physical image obtained) has inevitably changed. In this case we should consider whether the hash values of the single file have changed, and if so, which files have actually undergone changes due to the regular execution of the internal processes of the operating system. With the help of the capabilities of the acquisition software used, it was possible to extrapolate from the two physical images obtained a total amount of 9112 files. Only a very small percentage (less then 0.9% ) of these files, listed in Table 1, produce a different hash code.

It was also noted that the files that have undergone a change belong to a few different types: we have text files that can be traced back to the applications installed on your phone, database files, log files, and XML files. We can now analyze how these files can maintain links with data that are a possible source of evidence and for which the complete repeatability of the extraction process has to be demonstrated. Let us begin to delve into the nature of the first three files in the previous table, which, as can be seen from the path, are related to the messaging application WhatsApp, from which clearly emerge numerous sources of evidence. Within these text files, there is the sequence of messages exchanged with a given contact in the address book using the above application. The observation of a different hash at these files could create alarm, because it would lead us to believe that the extraction of evidence related to the messaging produced with this application does not have repeatability properties. In fact, a more in-depth analysis has shown that the aforementioned files do not belong to WhatsApp, in the sense that they are not files saved within the phone memory but rather are automatically generated by the extraction software when the reporting is created. The purpose of these files is to illustrate the content of chat conversations in a more intelligible format, but in reality—we insist—these files do not come from the internal memory of the phone. They have the same content in terms of address book contacts with which the messaging was exchanged and, having been produced at different times, they clearly have different hashes. In order to demonstrate the repeatability of the acquisition of digital evidence belonging to WhatsApp, it must therefore be demonstrated that the internal phone files belonging to that application have the same hashes. Let us start by making it clear that there is no single standard by which this application organizes these files within the phone’s memory: this organization depends on many factors, such as the operating system installed and whether or not there is a removable memory. In the case of this phone, which does not have removable memory, the application saves messaging data in the following database files:msgstore.db contains the encryption key with which messages sent from the phone are encrypted;wa.db in which messages properly encrypted with the above key are saved.

The study of repeatability must therefore focus exclusively on these files. By comparing the hash codes, we found that these hash codes are perfectly identical, therefore, we have sufficient reason to believe that the extraction of digital evidence attributable to the WhatsApp application is a perfectly repeatable operation in the perspective of a physical acquisition process. Let us now, instead, study the nature of the other files for which we have found an alteration of their hashes. We note that a particularly copious type of file, for which a hash variation has been found, is that with the .eml extension. These files are a “local” version of an e-mail message, so they can be traced to mail client applications. In the case of the phone examined, the only application present in this sense is the famous Gmail, application that allows access to Google domain mailboxes. For reasons of constitutional legislation, the issue of e-mail acquisition is rather thorny: not only because a different approach is needed depending on the virtual dimension in which the e-mail is located (e.g., saved locally because it is managed by a mail client program or saved in a webmail and therefore needs to be acquired by Internet forensics techniques), but, above all, because it alleges a violation of constitutional rights to secrecy of correspondence.

Returning to the analysis we are concerned with, those messages are obviously linked to an e-mail client app, since the telephone, at the time of acquisition, was without any connection to the Internet, yet that type of message has numerous hash alterations. Again, such considerations should not be alarmist, as we have found some rather interesting details from closer examination. In fact, some .eml files do not actually contain proper e-mail, but rather some only have a description of them: This is because metadata-related information such as the type of encoding used, the subject of the mail message in an encrypted format, the MIME extension for encoding multimedia attachments, etc. The cause of the hash variation is probably due to a different value presented from time to time at the date field: we note, first of all, the total absence of textual content that can be assimilated to a real e-mail message, but above all we found a variation exclusively of the said field. As we have already stressed several times, during the various mining activities the phone was kept in airplane mode, so it was not possible to produce other emails or alter those already existing.

Among the files with the .eml extension, it was also found that some of them reproduce the content of an email which, for obvious reasons, represents data validly that can be considered as computer evidence. In this case, we did not identify among these files, obtained during the two extractions of the different contents in the two versions, files such as to justify a variation of the respective hash code. As a result, we decided to analyze the metadata of these files, with particular reference to the values of the “creation date” and “last modified date” fields accessible from the properties window of the Windows environment. Further, this analysis, in fact, clarified our doubts, since the date of creation of the file did not coincide with the time moment when the email was sent/received, but rather coincided with the moment in which the mail was extrapolated from the binary file through the creation of the reporting. In other words, the change in the hash is not due to an alteration of the data, but rather to the algorithm with which Ufed Physical Analyzer produces, from the acquired image, the representation of the contents: for some of them, in fact, it simply reconstructs the file as extracted from the phone, in other cases, exploiting files traceable to some applications (such as in the case in under consideration those of e-mail) processes the contents of those files and generates new ad hoc files at the generation of the report. Indeed, it should not be forgotten that this instrument was designed to be invasive and not to alter, even accidentally, the analyzed devices, but certainly among properties there is no obligation to ensure the complete repeatability of all extracted data.

The files obtained from time to time have the same content, without altering the ‘substance of evidence’ represented by the content of the mail itself, so that we have sufficient information to infer, again, a repeatability of the acquisition process.

Another type of file, for which many variations have been found, is the file with the .xml extension. These files contain settings for applications, and it is from the applications that the content encapsulated in the various tags changes. Again, we cannot identify exactly what processes are responsible for managing these files, but we can be absolutely sure that the inevitable alterations to these files do not result in any pollution of the data that represent the sources of evidence. Just to give an example, we analyzed the contents of the phone-account-registar-state.xml file: within this document, it appears that a phone status is recorded at the interaction with a signal provided by a telephone operator which is the same as the one which owns the SIM inserted in the device. This file does not retain the same hash in the two acquisitions because the second version, obtained from the extraction, is not recorded in the <label> and <shor_description> tag of the telephone operator acronym.

Finally, the last type of file for which variations were observed is the database type. More precisely, .db files are managed by phone internal routines to store a set of information in a structured manner. Suffice it to say that, without the acquisition of root permissions, access to these files is inhibited as they may contain strictly confidential information that is essential to ensure the integrity of the phone. Using an sqlite browser you obtain the structure of the data saved in the offline.db file. In addition, you can see a series of tables that handle information related to the functioning of the phone, for example, the active processes (all identified with a set of text fields such as account_id, storage_id), the mapping of resources, etc. In this case, too, there is complete independence between the data that are managed by these files and the internal phone data that can be used as evidence; this independence is in fact the common factor of all the considerations made in this section. In other words, we have been able to demonstrate that the physical extraction processes, although they suffer from inevitable variations due to the normal operation of the phone—which, as stated above, must be kept on and restarted to allow the correct execution of the extraction—are, however, able to realize, in a perfectly repeatable manner, a forensic acquisition of data that can be used, as digital evidence, for investigative purposes.

### 5.2. Comparing File System Acquisition vs File System Acquisition

We can now analyze the results obtained with the two extracting file system processes conducted on the phone. Both processes returned 374 files of which only 30 detected a change in their hash codes (so in percentage terms only 8.02% of the total). We note from this analysis that essentially there are no different results to those obtained in the case of physical extraction processes. Files that vary as a result of the two capture processes are of the same type as those highlighted in the previous section, which is .db files (18 out of a total of 30) and .xml files (7 out of a total of 30).

Of all files, we decided to investigate in particular the nature of a database file for which a hash alteration was recorded: we’re talking about the calendar.db file. This database file is used by applications that function as a calendar or electronic calendar.

In the case of the device under consideration, the application that manages that file is Google Calendar. Confirmation of this link is also given by the analyzing the file through an Sqlite browser, thanks to which within the tables contained in the said file, we observe numerous records having, as the value of the field “accounts”, the domain address Gmail combined with that phone and with which is accessed to the play store. Analyzing the database file, we were unable to isolate any discrepancy between the contents present in the two versions of the database file, obtained from the two comparative extraction processes.

It is crucial to understand whether a change in the calendar.db file would somehow result in a pollution of the evidence sources extractable from the calendar application: the latter can be a valuable source of evidence as it is useful for checking, if necessary, what commitments and memoranda are recorded by the suspect. As a result, we decided to take advantage of the reporting generated by UFED Physical Analyzer, which also has a section dedicated to the memos registered in calendar applications on the phone. The comparison of the generated reporting allowed to recount the presence in the calendar application of 57 reminders, which were extracted in both acquisitions and listed in the related reporting in the same order. For these reasons, we cannot accurately identify the internal causes and processes of the phone that have touched, between an extraction and another, the aforementioned database file; We have shown, however, that the extraction of such data is, in fact, a repeatable process, since, as well as established case law expressed by the Supreme Court of Cassation, no alterations are observed that would result in a loss or alteration of the evidential content embodied in the extracted data.

As regards the remaining files, we can in fact draw the same conclusions as the previous acquisition, i.e., there is no alteration other than system files managed by both the applications and the internal processes of the operating system, which is why we have no reason to believe that the acquisition of those files that can constitute valid digital evidence, through a process of acquisition of the type we are considering, is also an activity that can be repeated.

### 5.3. Comparing Logical Acquisition vs. Logical Acquisition

Finally, in this section, we make the last comparison between acquisitions of the same type, analyzing the results obtained in two logical extractions.

The two extractions returned 388 files, of which only 18 (listed in Table 2) show a different hash value (4.63% of the total). As we can see, there are variations on files of the same type as the previous acquisitions, also finding a wide intersection with the files that in the previous processes had, between one activity and another, a different hash code.

Again, there are no modifications on the files that can be taken as evidence and managed directly by the user. Thus, it is well possible to consider, in the case of a logical extraction, the perfect repeatability of the process of acquiring the same files.

## 6. Conclusions

At the end of this paper, some reflections are made on the results presented. In the case of mobile devices such as, in this context, smartphones, the issue at stake, as we have seen, changes totally: not only do these devices have protective measures that prevent full access to the memory, but above all they do not all have the characteristics that allow a post-mortem analysis. The best practices, successfully tested in the world of computer forensics and the Law n.48/2008, which regulates the complex theme of cybercrime in Italy, have not found a full realization in the field of mobile forensics, given the impossibility of carrying out an acquisition activity that allows cloning, in its entirety, the internal memory of the phone through a perfectly repeatable process. Consequently, pending a common understanding of the choice to carry out a forensic analysis on a device under investigation by applying the regime of repeatable or non-repeatable tests, the Judicial Authority has frequently chosen the most prudent solution, that of the non-repeatable tests.

The question to be addressed is what, among the countless computer data that can be extrapolated from a phone, can validly constitute digital evidence. The results achieved are an additional element in favor of an orientation that tends to restrict only to data managed directly by the user the context in which to search for valid digital evidence and in relation to which the repeatability of the related acquisition process must be satisfied. It remains to be clarified the nature of those files that have seen a change in the hash code between acquisitions. In this regard, we have identified files that have different hash codes at two physical acquisitions, and it is precisely because of these files that these operations do not constitute perfectly repeatable acts. Regrettably, with regard to the Apple world, we cannot provide detailed reflections as these devices do not allow a physical acquisition of them, even with the tools provided to the judicial police. In the Android world, however, system files with hash variations are essentially groupable in three types:Database files with a .db extension;Text files, mostly with the .xml extension;Files of various kinds, such as images and texts reproducing emails, created at the same time as they were extrapolated from the compressed file produced by Ufed downstream of the acquisition, which is why, while exhibiting the same content, they were different from time to time, including Log files, which contain logs of system events that occurred on the device.

It is quite unusual that technical tests of an unrepeatable nature are carried out on the latter type of files. Thus, it is true that the contents of the log file are different depending on the moment in which they are analyzed (therefore the unrepeatability of the inspection), but if examination of such a file is considered essential for investigative purposes, then the same should be carried out following the path of urgent inspections according to article 354 [12] of the CPP and not that of the unrepeatable technical inspections under article 360 of the CPP since, in these circumstances, there is a need to ensure a source of evidence that would otherwise be lost. Finally, it should not be forgotten that the national legal scenario seems to be more in line with our considerations, as it seems that the Legislator intends to assess the repeatability of the inspection in relation to the ‘computer data’ and not, as in the past, to the whole ‘computer support’.

## Figures and Tables

**Table 1 sensors-22-07096-t001:** List of the file extracted from the smartphone under analysis with the hash code modified. All of them have been created by the user.

File Name	File Name
S4/chats/WhatsApp/chat-1.txt	S4/files/Database/phenotype.db
S4/chats/WhatsApp/chat-2.txt	S4/files/Database/rut.db
S4/chats/WhatsApp/chat-3.txt	S4/files/Database/scheduler
S4/email/racuomo88@gmaiL.com/Inbox/mes-88.eml	S4/files/Database/scheduler_logging_store.db
S4/email/racuomo88@gmaiL.com/Inbox/mes-84.eml	S4/files/Database/google_app_measurement_locaL.db
S4/email/racuomo88@gmaiL.com/Inbox/mes-90.eml	S4/files/Database/google_app_measurement_local_1.db
S4/email/racuomo88@gmaiL.com/Inbox/mes-96.eml	S4/files/Database/offline.db
S4/email/racuomo88@gmaiL.com/Inbox/mes-99.eml	S4/files/Text/0.xml
S4/email/racuomo88@gmaiL.com/Inbox,/mes-16.eml	S4/files/Text/MT_shared_fref.xml
S4/email/racuomo88@gmaiL.com/Inbox,/mes-74.eml	S4/files/Text/settingsprovider.txt
S4/email/racuomo88@gmaiL.com/Inbox,/mes-76.eml	S4/files/Text/setupwizard.txt
S4/email/racuomo88@gmaiL.com/Inbox,/mes-83.eml	S4/files/Text/package-restrictions.xml
S4/email/racuomo88@gmaiL.com/Inbox,/mes-85.eml	S4/files/Text/packages.xml
S4/email/racuomo88@gmaiL.com/Nativo/mes-103.eml	S4/files/Text/pending.xml
S4/email/racuomo88@gmaiL.com/Nativo/mes-104.eml	S4/files/Text/jobs.xml
S4/email/racuomo88@gmaiL.com/Nativo/mes-105.eml	S4/files/Text/LOCATION_REPORTING.xml
S4/email/racuomo88@gmaiL.com/Nativo/mes-106.eml	S4/files/Text/salter_pref.xml
S4/email/racuomo88@gmaiL.com/Nativo/mes-107.eml	S4/files/Text/BEACON_STATE.xml
S4/email/racuomo88@gmaiL.com/Nativo/mes-108.eml	S4/files/Text/com.google.android.gms.measurement.prefs.xml
S4/email/racuomo88@gmaiL.com/Nativo/mes-109.eml	S4/files/Text/Log_2.txt
S4/email/racuomo88@gmaiL.com/Nativo/mes-110.eml	S4/files/Text/finsky.xml
S4/email/racuomo88@gmaiL.com/Nativo/mes-111.eml	S4/files/Text/flipboard_settings.xml
S4/email/racuomo88@gmaiL.com/Nativo/mes-112.eml	S4/files/Text/abt-persistence-service1.log
S4/email/racuomo88@gmaiL.com/Nativo/mes-113.eml	S4/files/Text/omacp0.log
S4/email/racuomo88@gmaiL.com/Nativo/mes-114.eml	S4/files/Text/GmsBackupTransport.backupScheduler.xml
S4/email/racuomo88@gmaiL.com/Nativo/mes-115.eml	S4/files/Text/GmsBackupTransport.restoreScheduler.xml
S4/email/racuomo88@gmaiL.com/Nativo/mes-116.eml	S4/files/Text/com.google.android.gms.measurement.prefs_2.xml
S4/email/racuomo88@gmaiL.com/Nativo/mes-117.eml	S4/files/Text/com.google.android.gms.measurement.prefs_4.xml
S4/email/racuomo88@gmaiL.com/Nativo/mes-118.eml	S4/files/Text/com.jrtstudio.AnotherMusicPlayer_preferences.xml
S4/email/racuomo88@gmaiL.com/Nativo/mes-119.eml	S4/files/Text/com.mopub.settings.identifier.xml
S4/email/racuomo88@gmaiL.com/Sent/mes-53.eml	S4/files/Text/tzdrm.log
S4/email/racuomo88@gmaiL.com/Sent/mes-69.eml	S4/files/Text/uiderrors.txt
S4/email/racuomo88@gmaiL.com/Sent/mes-66.eml	S4/files/Text/sk.log
S4/email/racuomo88@gmaiL.com/Sent/mes-92.eml	S4/files/Text/appops.xml
S4/files/Database/accounts.db	S4/files/Text/audit.log
S4/file/Database/launcher.db	S4/files/Text/phone-account-registrar-state.xml
S4/files/Database/enterprise.db	S4/files/Text/event_generator.xml
S4/files/Database/config.db	S4/files/Text/power_off_reset_reason.txt
S4/files/Database/telephony.db	S4/gps/viaggi.kml
S4/files/Database/phenotype_1.db	S4/journeys/Runtastic/Raffaele Cuomo/journey-1.txt

**Table 2 sensors-22-07096-t002:** List of the extracted files that have been modified (with a different hash code. (12 are .db files and 6 are .xml files).

File Name
calendar.db
com.google.android.Libraries.youtube.net.delayedevents.DelayedEventStore
counters.db
offline.db
phenotype.db
scheduler
scheduler_logging_store.db
icingcorpora.db
localappstate.db
frosting.db
gmm_storage.db
google_conversion_tracking.db
finsky.xml
flipboard_settings.xml
install_queue.xml
settings_preference.xml
youtube.xml
event_generator.xml

## Data Availability

Not applicable.

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
