# Peer review of "Mobile Forensics: Repeatable and Non-Repeatable Technical Assessments"

_sensors, 2022, doi:10.3390/s22187096_

Round 1

Reviewer 1 Report

The use of abbreviations in the paper is not standardized; the first time it is used to give the full name. It is fine to use abbreviations for all subsequent times.

The case in the paper is also not standardized. Some statements are lowercase as sentence starters.

This paper does not look like a research paper. If it is a review paper, various figures and tables should be given and conclusions drawn accordingly.

Author Response

Reply to Referee’s criticism, Manuscript # sensors-1813475

We thank the referee for his suggestions.

As reported by the attached ChangeLog.pdf, many interventions have been made to improve the readability of the work and the entire manuscript has been revised by a native English speaker expert and the quality of the paper has greatly improved.

Referee says :

The use of abbreviations in the paper is not standardized; the first time it is used to give the full name. It is fine to use abbreviations for all subsequent times.

Our reply :

We corrected this mechanical deficiency all over the manuscript. We introduce the latex package acronym to handle in homogeneous way all the acronyms.

Referee says :

The case in the paper is also not standardized. Some statements are lowercase as sentence starters.

Our reply :

We have corrected the text to use the proper case for the letters all over.

Referee says ;

This paper does not look like a research paper. If it is a review paper, various figures and tables should be given and conclusions drawn accordingly.

Our reply :

We respectfully observe that the material presented in sections 4 and 5 reports mostly original work, information, and processes that are extracted from authors work and experience. The presentation is original and no parts of it have been previously published. Also, Table 1 contains substantial and quantitative information on files and data employed for our endeavour.

Reviewer 2 Report

- Starting a sentence with However, no matter that it is referenced - is not usual and should be improved

- Airplane mode is (sometimes) not sufficient mode for isolating device because in many mobile phones / smartphones the Wi-Fi is still active

- "it possible to carry out penetrating and effective analyses capable of returning robust evidence from the legal point of view" is referenced sentence and shouldn't be C/P as authors sentences

- "context of I-Device" according to my knowledge, not used often and should be changed

- in Conclusion there should not be referenced sentences

Author Response

Reply to Referee’s criticism, Manuscript # sensors-1813475

We thank the referee for his suggestions.

As reported by the attached ChangeLog.pdf, many interventions have been made to improve the readability of the work and the entire manuscript has been revised by a native English speaker expert and the quality of the paper has greatly improved.

Referee says :

Starting a sentence with However, no matter that it is referenced - is not usual and should be improved

Our reply:

We have corrected this syntax problem all over the manuscript.

Referee says :

Airplane mode is (sometimes) not sufficient mode for isolating device because in many mobile phones / smartphones the Wi-Fi is still active

Our reply :

Referee is right and we thank for the suggestion. We have better specified/clarified now at line 43 :

“Airplane mode (switching off Wi-Fi and any other communication channel like Bluetooth as well”)

Referee says :

“It possible to carry out penetrating and effective analyses capable of returning robust evidence from the legal point of view" is referenced sentence and should not be C/P as authors sentences

Our reply :

The entire sentence was written by the authors and the reference was deleted

Referee says :

"context of I-Device" according to my knowledge, not used often and should be changed

Our reply :

We are sorry for having used an improper term from a technical (slang) dictionary. We have changed I-Device with “mobile devices released by Apple” as reported in the attached file ChangeLog.pdfon line 148.

Referee says :

- in Conclusion there should not be referenced sentences

Our reply :

All the references have been removed from the section “Conclusions”.

Round 2

Reviewer 1 Report

 I have no more comments.